# Genetic and Genomic Epidemiology of Stroke in People of African Ancestry

**DOI:** 10.3390/genes12111825

**Published:** 2021-11-19

**Authors:** Savvina Prapiadou, Stacie L. Demel, Hyacinth I. Hyacinth

**Affiliations:** 1Department of Medicine, University of Patras School of Medicine, 26223 Patras, Greece; sprapiadou@gmail.com; 2Department of Neurology and Rehabilitation Medicine, University of Cincinnati College of Medicine, Cincinnati, OH 45221, USA; demelse@ucmail.uc.edu

**Keywords:** stroke genetics, GWAS, Africans, African Americans, stroke ethnic disparities

## Abstract

Stroke is one of the leading causes of disability and death worldwide and places a significant burden on healthcare systems. There are significant racial/ethnic differences in the incidence, subtype, and prognosis of stroke, between people of European and African ancestry, of which only about 50% can be explained by traditional stroke risk facts. However, only a small number of genetic studies include individuals of African descent, leaving many gaps in our understanding of stroke genetics among this population. This review article highlights the need for and significance of including African-ancestry individuals in stroke genetic studies and points to the efforts that have been made towards this direction. Additionally, we discuss the caveats, opportunities, and next steps in African stroke genetics—a field still in its infancy but with great potential for expanding our understanding of stroke biology and for developing new therapeutic strategies.

## 1. Introduction

Stroke is a complex and heterogeneous disease. Risk of stroke involves the interplay between genetic, epigenetic, and environmental factors. Stroke is the second leading cause of death and disability worldwide with over 13 million new strokes and 5.5 million deaths recorded in 2016 [1]. Although the overall incidence and therefore burden of stroke in developed countries decreased by 42% in the period 1970–2008, it increased by more than 100% in low-income countries and in developing countries, where there is a disproportionate increase among individuals of African ancestry [2]. Additionally, community-based studies reported an age-standardized stroke incidence rate up to 316 per 100,000 population in some populations in Africa, placing a significant burden on the already struggling healthcare systems [3,4,5,6]. Annually, it is estimated that 3.2 million Africans will have their first stroke [5]. Additionally, African Americans are 50% more likely to have a stroke and 70% more likely to die from a stroke compared to their non-Hispanic White (NHW) counterparts [7]. 

The incidence of stroke is twice as high in African Americans compared to Americans of European ancestry. This disparity in incidence is more pronounced for ages 45 to 54 years (Black/white Incidence rate ratio (IRR) = 4.02) and declines with increasing age (IRR for individuals ≥85 years of age = 0.86) [8,9,10] possibly suggesting a role for genetics since gene loading is expected to occur at an earlier age. There are also notable differences in the distribution of stroke types and subtypes between individuals of African ancestry and those of European ancestry. Findings from the INTERSTROKE study indicate that among Africans in Africa, 34% of all strokes were hemorrhagic and 66% were ischemic, as opposed to 9% and 91% respectively among mixed populations in high-income countries [11]. The South London Ethnicity and Stroke Study (SLESS) study also found a higher incidence of cerebral small vessel disease (odds ratio [OR]: 2.74) and less extracranial large vessel atherosclerosis (OR: 0.59) and cardioembolic stroke (OR:0.61) among stroke patients of African ancestry living in the United Kingdom (UK), compared to UK patients of European ancestry [12]. Additionally, the mean age of stroke occurrence is 9 years younger (57 compared to 66 years) in populations in Africa, compared to combined populations in high-income countries [5]. Finally, the South London Stroke Registry study (multiethnic population within London) found that black stroke survivors had worse cognitive outcomes after stroke [13]. 

The reasons for the above racial disparities in stroke occurrence and outcomes are multifaceted and remain largely unknown. It is hypothesized that differences in the rates of stroke risk factors, such as uncontrolled hypertension, might be a possible culprit. However, data from the biracial Reasons for Geographic and Racial Differences in Stroke (REGARDS) study found that only 50% of the increased stroke incidence reported in blacks could be explained by racial peculiarities in the prevalence of traditional risk factors such as the Framingham risk factors or by socioeconomic status, income and education [8]. Additional data from REGARDS, showed that uncontrolled hypertension might also be a factor, however, the reason for the lack of control was not clear from the study, given that Blacks were more likely to be aware of their blood pressure and also be on treatment or on multiple blood pressure medications compared to non-Hispanic Whites [14]. Interestingly, African Americans have twice the level of strokes for the same level of systolic blood pressure (SBP) as their White counterparts [15]. Given these findings and the estimated high heritability of stroke it is likely that at least some if not all of the other half of the excess in incidence, morbidity and mortality, could be attributed to genetic and non-traditional risks. 

Genetic studies of stroke in populations of African ancestry are crucial not only for unveiling the reasons for the pronounced ethnic disparities in stroke incidence, subtype distribution and outcome but also for elucidating the genetic and biological mechanisms of stroke globally and even eventually moving towards developing precision medicine for stroke. Unfortunately, most genetic stroke studies involve primarily populations of European ancestry, although a few also include African Americans and only a couple focus on continental Africans. 

It is important to recognize the current state of stroke genetics in African-ancestry populations, the recent and upcoming advances and the work that still needs to be done. In this article we will review the findings of recent Genome-Wide Association Studies (GWAS), Whole-Exome Sequencing (WES) studies and candidate gene studies that have included individuals of African Ancestry (living in the US, Europe and Africa), focusing on their advantages and shortcomings. We will also examine the existing knowledge gaps, challenges, opportunities and then discuss some next steps in the efforts to uncover the genetic underpinnings of the distinctive epidemiology and genetic epidemiology of stroke in these populations. 

## 2. Materials and Methods

A systematic literature review was conducted in accordance with the PRISMA guidelines using the PubMed and Google Scholar databases. Combinations of keywords used to generate the list of articles include: “African/African American/ Black stroke genetics”, “Stroke GWAS Africans /African Americans/Blacks”, “Ischemic stroke genetics Africans/African Americans/Blacks”, “Candidate gene studies stroke Africans/African Americans/Blacks”, “Exome Sequencing stroke Africans/African Americans/Blacks” and “Hemorrhagic stroke genetics Africans/African Americans/Blacks”. Please note that, in order to search more broadly, search phrases were not included in quotation marks and each keyword referring to individuals of African ancestry (“Africans, African Americans, Blacks”) was included in a separate search. Only studies published between January 2000 and May 2021 were selected. After eliminating duplicates, the studies were then characterized as GWAS, WES studies, candidate-gene, case-control, population-based and meta-analyses. Articles that were not written in English, did not pertain to studies of individuals of African ancestry or to stroke genetic studies and study protocols were excluded from the review (Figure 1). Non-specific study types, studies of other populations (reference cohorts) and those referenced in the introduction section of the article are referred to as “others”.

## 3. Results

### 3.1. The African Genetic Architecture and Stroke Genetics

People of African ancestry, as a result of their longer history, have greater genetic diversity, higher levels of admixture and an intricate population substructure pattern compared to other ancestral groups. Because of this, individuals of African ancestry have the lowest levels of linkage disequilibrium and the smallest haplotype blocks compared to individuals of European ancestry. Due to the need to fight off certain pathogens and diseases, populations in Africa and therefore individuals of African Ancestry have developed genetic adaptations that have resulted in the presence of distinct allele frequencies, not seen in or among other ancestral groups/populations. Some of these once beneficial alleles can increase the risk of disease in certain environments. Examples include the APOE ε4 and APOL1 gene variants as well as sickle cell disease (SCD) all of which once provided a survival advantage (for instance against malaria in the case of the sickle cell trait (SCT) and against Trypanosoma brucei infections in the case of *APOL1*) but are now shown to also increase susceptibility to non-communicable diseases [16,17].

The higher burden of stroke in people of African Ancestry might be partially attributable to differences in ancestry and genetic risk. Data from the SLESS study found that single nucleotide polymorphism (SNP) heritability of stroke is higher among African-ancestry populations compared with individuals of European ancestry, suggesting a stronger influence of genetics in stroke risk among this population [18]. The presence of higher stroke heritability, increased genetic variation, and greater prevalence of stroke found in African populations serve as advantages to help uncover the unknown genetic and genomic contributors to stroke risk. 

### 3.2. Sickle Cell Anemia and Stroke

SCD has been known to significantly increase the risk for overt ischemic stroke, silent cerebral infarcts, and hemorrhagic stroke [19]. More specifically, a child with SCD is 333 times more likely to suffer from stroke compared to a healthy child without SCD. Overall 24% of patients with SCD present with a stroke by the time they are 45 years old [20] and an additional 17–22% will have silent cerebral infarctions only seen on magnetic resonance imaging [21]. Patients have an increased risk of ischemic stroke earlier and hemorrhagic stroke later in their lifetime. The reasons for the increased incidence of stroke among patients with SCD are thought to be related to the anemia/hypoxemia caused by the disease as well as endothelial, red blood cell and platelet-related processes and an excess of inflammatory factors [22].

The role of the Sickle Cell Trait (SCT) (heterozygous state of the sickle cell mutation) in stroke risk is not as straightforward. An older, longitudinal single-cohort population study had suggested a link between SCT and ischemic stroke [23]. However, a recent meta-analysis of four large prospective population studies which included 19,464 African Americans concluded that individuals with sickle cell trait did not have significantly increased stroke incidence when compared to controls [24]. The differences between the above two studies include sample size, age range of the participants and the covariates that were controlled for in the logistic regression analysis. Further, well-designed studies with large numbers of African and African American participants and including a long follow-up of patients are needed to establish the link, if any, between the sickle cell trait and stroke incidence, including stroke subtypes [23,24]. 

### 3.3. Candidate Gene Studies

One way to uncover the ethnic differences in stroke prevalence is through targeted analyses of specific genes and alleles. One candidate allele which is found more frequently in African populations and is associated with several different phenotypes is the *APOL1* risk allele, which has two variants [25]. In an analysis of 10,605 black participants without renal disease from the REGARDS Study, possession of two *APOL1* high-risk variants significantly increased the risk for ischemic and small vessel disease-related stroke [26]. Other examples include the *Low-density lipoprotein receptor related proteins-1* and *6* variants which have been long linked to ischemic stroke. In a case-control study of the Ischemic Stroke Genetics Study (ISGS) cohort, the *LRP1* rs11172113 variant was associated with stroke among African Americans without a replication of that association in the non-Hispanic White cohort [27]. Other candidate genes include *Interleukin-6 (IL-6)* and *CDKN2A/CDKN2B* which have been implicated as sources of the racial disparities in stroke incidence. A candidate gene analysis of indigenous West African participants in the Stroke Investigative Research and Education Network (SIREN) Study revealed two genetic polymorphisms; rs1800796 in *IL6*, and rs2383207 in *CDKN2A/CDKN2B*, both of which were associated with increased stroke incidence in male participants [28]. Another study of the SIREN cohort concluded that SNPs in *APOL1*, *CDKN2A/CDKN2B*, and *HDAC9* genes were significantly associated with small vessel disease ischemic stroke risk [29]. Additionally, the intron 4c allele in the *NOS3* gene was linked with large artery ischemic stroke among African Americans in a small sample of 377 patients with ischemic stroke [30]. Finally, an analysis of the population-based, biracial Stroke Prevention in Young Women case-control study found two SNPs in the *NOS3* gene associated with ischemic stroke in the black women but not in white women [31]. A review of the candidate genes identified to be possible contributors to the racial disparities in stroke can be found in Table 1. 

In contrast, other alleles that have been established to influence stroke risk in non-Hispanic White individuals do not seem to be genetic contributors to stroke risk in Blacks. For example, polymorphisms in the Arachidonate 5-lipoxygenase activating protein (*ALOX5AP*) gene were associated with ischemic stroke in White Americans but not in African Americans [46]. In addition, the *APOE* locus which has been associated with lobar and deep Intracerebral Hemorrhage (ICH) in European populations [47] does not have an independent effect in Hispanics or individuals of African Ancestry, according to a large meta-analysis of 11 hospital and population-based studies in Europe and the US, although other multiethnic studies have reported conflicting results regarding the role of the specific alleles in hemorrhagic stroke risk across different ethnicities [48,49]. 

Collectively, the results of the above candidate-gene studies suggest some shared and some distinctive mechanisms of stroke development between ancestral groups. More studies targeting specific polymorphisms thought to play a role in stroke pathophysiology and involving a larger number but more ancestrally diverse populations, with adequate power for analysis by stroke subtypes, are needed to unravel the role of specific genes and polymorphisms in stroke incidence across the globe. 

### 3.4. GWAS and WES Analyses of African and African American Stroke Patients

GWAS, as opposed to candidate gene studies, have been successful in identifying novel variant-trait associations and can provide new insights into ethnic variation of complex traits such as stroke. Unfortunately, the results of GWAS and WES studies to date have been primarily driven by individuals of European ancestry. The first large-scale total stroke GWAS of African American individuals came from the Consortium of Minority Population genome-wide Association Studies of Stroke (COMPASS) collaboration. The analysis, which included 14,746 African Americans, found a novel SNP associated with total stroke in the 15q21.3 locus which is located near the *Aquaporin 9 gene (AQP9)*, the *aldehyde dehydrogenase 1 family, member A2 (ALDH1A2)* and *hepatic lipase (LIPC)* genes. This region has been linked with cerebral energy metabolism, brain ischemia, lipid levels and hypertension. 18 additional SNPs showed suggestive association and multiple loci had the same direction of effect in stroke risk as in European ancestry populations, pointing to common pathogenetic mechanisms of stroke across both ancestry groups [33]. A recent GWAS meta-analysis of 22,000 African Americans in the COMPASS cohort revealed a novel polymorphism near the *HNF1A* gene that was significantly associated with stroke and 29 new suggestive variants, some of which were also replicated in European cohorts [9].

Another GWAS of Africans and African Caribbeans in the SLESS study also showed that SNPs associated with ischemic stroke in Europeans had a similar impact in African populations [18]. Additionally, the genetics of early-onset stroke also seem to be influenced by ethnicity as demonstrated in an exome array analysis of the UMD-GEOS Study (University of Maryland-Genetics of Early-Onset Stroke) which identified mostly different SNP top hits in the White than in the Black population included in the study. One notable exception was a variant in the *VWDE* gene encoding von Willebrand factor D and EGF-domain–containing protein which was associated with early-onset stroke in both white and black individuals [50]. An overview of all the significant genetic stroke studies including African-ancestry participants can be found in Table 2. In summary, while some recent GWAS of African-ancestry individuals have identified suggestive and causative variants associated with total stroke, most still lack adequate power to stratify by stroke subtype (which is important given the heterogeneous pathophysiology between the different subtypes) or to even identify additional or less common variants with impact on susceptibility and/or outcome. Future large, multiethnic GWAS studies will be crucial in order to explain the different etiological mechanisms of stroke, stroke types as well as ischemic stroke subtypes in each ethnic population. Such studies will need to be adequately powered for ancestry-specific analysis, and not just transethnic analysis.

## 4. Discussion

### 4.1. Challenges 

Including African-ancestry individuals in stroke genetic studies presents several challenges. Firstly, the lack of reference panels that contain common variation of African-descent populations makes it difficult to characterize and as well as consider the genetic differences between populations. Secondly, the absence of an array which includes SNPs found in African populations makes SNP imputation a challenging task [18]. Thirdly, the higher genetic diversity of African-descent populations means that larger sample sizes might be needed in general to find allele-trait associations and introduces the possibility of false positive findings attributable to population stratification [9], or false negative findings due to small sample sizes. Finally, there is a need for better ancestry inference algorithms from whole genome sequencing data to identify each individual’s specific ancestry across the diverse African populations [16]. The above technical shortcomings place significant obstacles in the path towards making genetic stroke studies inclusive and diverse. 

### 4.2. Opportunities

Although including African ancestry populations in stroke genetics can seem a challenging task, the benefits and opportunities of such studies are immense and can have important global implications since all humans descended from Africa. Replication of European-descent population risk variants in African descent populations increases the generalizability and scope of the findings by indicating a true causal relationship between the candidate variant and the trait [51]. Additionally, the distinctive genetic characteristics of individuals of African ancestry such as their low linkage disequilibrium makes it easier to pinpoint the specific causal variants responsible for the trait through fine mapping of the GWAS findings [52]. Furthermore, including transethnic populations can reveal pathogenetic mechanisms that would not otherwise be found in European ancestry-only analyses. Finally, due to the higher genetic heritability of stroke among African ancestry populations, identifying variants involved in stroke risk could require a smaller sample size than for European populations [18]. The knowledge derived from African stroke genetic studies could then be used to develop accurate stroke risk stratification techniques, precise biomarkers and personalized medicine approaches for better stroke prevention and treatment, narrowing the persistent racial gap in stroke incidence and mortality and lessening the overall burden of stroke. 

### 4.3. Future Directions

The important role of GWAS in discovering the biological pathways that lead to diseases, including stroke, has been highlighted. Although the number of GWAS performed increased more than 6-fold from 2009 to 2016, the number of African ancestry participants increased only by 2.5% [9]. These disappointing numbers highlight the failure of GWAS studies to include diverse populations which can only be resolved through a multifaceted approach. One effort that will aid and is currently moving research in this direction, is The Human Hereditary and Health in African Consortium which was established by the National Institutes of Health (NIH) and the Wellcome Trust to develop the necessary infrastructure and expertise for genomic research in Africa. One important goal of the consortium is to create a pan-African genotype array that captures most of the genetic variants found in African ancestry populations [16,53]. Further, we need to invest in African scientists and doctors through government and industry funded efforts, geared toward building the necessary foundations and capacity needed for a more uniform/accurate definition of different stroke and related phenotypes, as well as in the use and interpretation of genetic information. 

Efforts to include African populations specifically in stroke studies include SIREN which includes the largest population of Continental Africans with stroke and the COMPASS consortium which was founded to identify susceptibility loci for stroke in minority populations. Additional examples of studies that seek to elucidate the ethnic discrepancies in stroke incidence are The Genetics of Early Onset Stroke (GEOS) Study which includes a large number of African American stroke patients and the SLESS which includes a UK African ancestry population. The above cohorts would provide adequate statistical power for identifying the ancestry-specific genetic component of each of the stroke subtypes [18]. 

As we look ahead, it will become important to perform gene and pathway-based analyses of diverse cohorts and to combine genomics with functional-omics such as proteomics, metabolomics, transcriptomics and epigenomics. This information could help us understand the complex pathophysiology of stroke and to uncover the complex gene-gene and gene-environment interactions responsible for the ethnic disparities in stroke risk [52]. Based on the results of the above studies we can develop precise biomarkers for stroke prediction, diagnosis and prognosis and target biological pathways with new and existing drugs personalized for each patient, regardless of ethnicity.

This systematic review explores the few efforts that have been made to better understand the distinct genetic and genomic underpinnings of stroke in individuals of African ancestry, an ethnic group that is disproportionately affected by stroke. These studies have uncovered points of divergence and convergence of stroke pathogenesis between different ethnicities and populations. By discussing this understudied, yet highly significant topic, we hope to emphasize the need for further, more focused research that will improve our understanding of stroke etiology and stroke management in African ancestry populations. 

## Figures and Tables

**Figure 1 genes-12-01825-f001:**
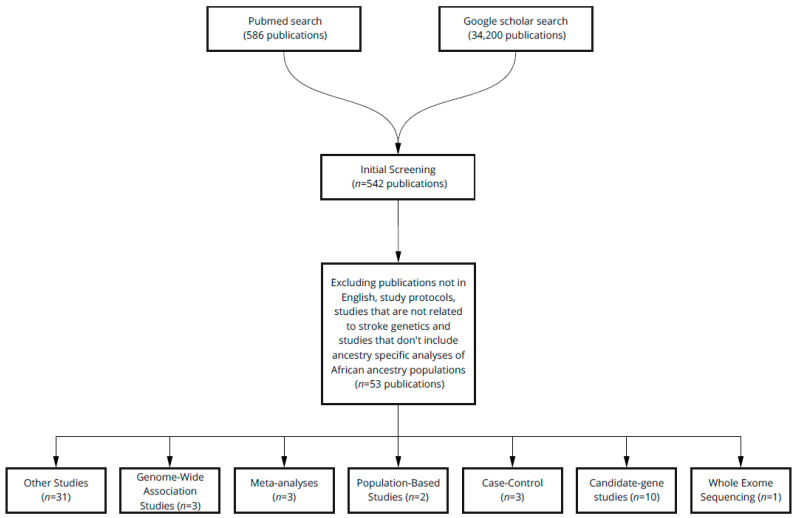
Publication selection process. The number of studies included in each step of the filtering process is stated and the criteria for excluding papers from the review are listed. The final number of studies (*n* = 53) is further classified as one of seven different categories depending on study type.

**Table 1 genes-12-01825-t001:** Genes and alleles implicated in ethnic stroke discrepancies.

Candidate Gene Variants	Primary and Replication Cohorts of Individuals of African Ancestry	Population Studied	Type of Stroke	Replication in Other Populations
Sickle Cell Trait	3497 individuals (223 with SCT, 401 with incident stroke) [23]	African Americans	Total stroke	None
	19,464 individuals (1520 with SCT, 620 with ischemic stroke) [24]	African Americans	Ischemic stroke	
*LRP1*	161 stroke patients, 116 controls from the Ischemic Stroke Genetics Study (ISGS); association with the rs11172113 variant [27]	African Americans	Ischemic stroke	None
*Apolipoprotein L1 (APOL1)*	154 cases and 483 stroke–free controls from the SIREN study; association with the rs73885319 variant [29]	Indigenous West Africans	Small vessel disease (SVD) ischemic stroke	None
	10,605 individuals from the REGARDS cohort; 1346 participants with high-risk *APOL1* genotypes [26]	African-Americans		
*CDKN2A/CDKN2B*	154 cases with stroke and 483 stroke–free controls from the SIREN study; association with the rs2383207 variant [29].	Indigenous West Africans	SVD ischemic stroke	3964 individuals of European ancestry; association with Atherosclerotic stroke [32]
	429 cases and 483 controls also from the SIREN study; association found in men but not in women [28].	Indigenous West Africans	Ischemic stroke	
*HDAC9*	154 cases with SVD ischemic stroke and 483 stroke–free controls from the SIREN study; association with the rs28688791 variant [29]. 1365 ischemic and 1592 total stroke cases from the COMPASS cohort; nominal association with stroke [33].	Indigenous West Africans	SVD ischemic stroke	2356 stroke patients and 3420 healthy controls of Chinese ancestry; association of rs2107595 with ischemic stroke [34].
		African Americans	Ischemic and Total stroke	3548 cases and 5972 controls of European ancestry. Additional replication cohort of 735 cases and 28,583 controls; association with the rs11984041 variant with ischemic stroke [35].
*Interleukin-6 (IL6)*	429 cases with SVD ischemic stroke and 483 stroke–free controls from the SIREN study; association with the rs1800796 variant [28]. 224 cases of ischemic stroke and 211 control subjects; association with SNP rs2069832 [36].	Indigenous West Africans	Ischemic stroke	Associations between *IL6* polymorphisms and stroke in American Non-Hispanic Whites (*n* = 2905) [37], Japanese (1141 stroke cases and 2010 controls
		African Americans	IschemicIschemicIschemic stroke	Ref. [38], Chinese populations (748 ischemic stroke patients, 748 controls;622 participants) [39,40] and a Northern Indian population (250 ischemic stroke patients and 250 controls) [41]
*NOS3*	377 patients and 502 controls; association with the intron 4c allele [30].	African Americans	Ischemic stroke	Association of the 27-bp repeat polymorphism in *ecNOS* gene with ischemic stroke in Chinese patients (364 patients with ischemic stroke and 516 control subjects) [42].
	Promoter variants in *NOS3* associated with ischemic stroke in black women [31].	African Americans	SVD ischemic stroke	The intron 4ab insertion/deletion genotype was linked with lacunar infarction in White patients (300 patients with SVD stroke and 600 community controls) [43].
*PITX2*	14,746 participants (1365 ischemic stroke and 1592 total stroke) from the COMPASS cohort [33].	African Americans	Ischemic stroke	Polymorphisms in the *PITX2* gene have been associated with various ischemic stroke subtypes in a European-ancestry population (12,389 individuals with ischaemic stroke and 62,004 controls) [44].
*ZFHX3*	14,746 participants (1365 ischemic stroke and 1592 total stroke) from the COMPASS cohort [33].	African Americans	Ischemic stroke	Variants in *ZFHX3* have been linked with ischemic stroke in European-ancestry populations (2224 cases and 2583 control subjects) [45].

**Table 2 genes-12-01825-t002:** Genetic Stroke studies including individuals of African Ancestry.

Name of Study	Type of Study	First Author	Date Published	Number of Individuals Included	Population Studied	Key Findings
Low density lipoprotein receptor related protein-1 and 6 gene variants and ischemic stroke risk [27].	prospective, multicenter genetic association study	Harriott et al.	08/2015	*Ν* = 1030	Caucasians (434 stroke patients, 319 controls) African Americans (161 stroke patients, 116 controls)	Among the caucasian participants 5 *LRP6* variants were found to be protective of ischemic stroke while among African Americans one variant in *LRP1* was found to increase stroke risk
Meta-analysis of genome-wide association studies identifies genetic risk factors for stroke in African Americans [33].	Meta-analysis	Carty et al.	08/2015	*N* = 14,746	African Americans (1365 ischemic and 1592 total stroke cases)	Association of the 15q21.3 locus with total stroke. Additionally, 18 variants were nominally associated with ischemic or total stroke. Replication of loci in *PITX2, HDAC9, CDKN2A/CDKN2B* and *ZFHX3* found to be associated with ischemic stroke in European-ancestry populations
Genome-Wide Association Study Meta-Analysis of Stroke in 22,000 Individuals of African Descent Identifies Novel Associations With Stroke [9].	Genome-wide association study	Keene et al.	07/2020	*N* = 22,000	African- Americans (3734 cases, 18, 317 controls) from 13 cohorts.	Identification of one SNP near the *HNF1A* gene that was significantly associated with stroke and 29 variants with suggestive association. Most of the variants have also been found to be associated with stroke in other populations.
Genetics of stroke in a UK African ancestry case-control study [18].	Case-control	Traylor et al.	04/2017	*N* = 1785	Africans and African Caribbeans (917 stroke cases, 868 age-matched controls from the UK)	The results showed a high heritability of stroke risk in the African ancestry populations. SNPs associated with ischemic stroke in Europeans share the same direction of effect in African ancestry populations.
Exome Array Analysis of Early-Onset Ischemic Stroke [50].	Whole Exome Sequencing	Jaworek et al.	9/2020	*N* = 1449	African Americans (723 controls and 721 cases)	*NAT10* was associated with small-vessel stroke. Several pathways related to neurotransmitter, neurodevelopmental, notch-signaling, and lipid/glucose metabolism were also identified to be related to stroke pathogenesis

## Data Availability

Not applicable.

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
