# Peer review of "Genetic and Genomic Epidemiology of Stroke in People of African Ancestry"

_genes, 2021, doi:10.3390/genes12111825_

Round 1
Reviewer 1 Report
Overall a good paper on an topic of high relevance, yet not much discussed.
Author Response
Thank you for your time and feedback
Reviewer 2 Report
This review article discussed the epidemiology of stroke in African-ancestry individuals. Authors well summarized the underlying genetic and genomic studies of stroke in African-ancestry subjects through the systematic literature review using the PubMed and Google Scholar databases. I think this manuscript is of interest and well written. Overall, this review will be very informative for readers, particularly who are interested the significant racial/ethnic differences in the incidence of stroke. There are a few concerns that should be addressed.
1. In Figure 1, quality of figure including letter size and font is poor. It should be revised. There is only figure title. The figure legend describing the flow diagram in detail should be added.
2. All gene names in the text and Tables should be presented by the italic letter.
3. The formats including letter font, line thickness, and space between the lines in Tables 1 and 2 should be revised.
4. Full description is required for the abbreviated terminologies that appear first time: eg., SBP (line 66), SNP (line 122), and NIH (line 270).
5. In the last part of the “Discussion” section, the summarized conclusions including the significance and key points of this review are needed.
Author Response
Response to Reviewer 2 Comments
Point 1: In Figure 1, quality of figure including letter size and font is poor. It should be revised. There is only figure title. The figure legend describing the flow diagram in detail should be added.
Response 1: A better quality image of the figure was added, and the font of the letters was changed. A description of the flow diagram was included which we hope will be useful to the readers.
Point 2: All gene names in the text and Tables should be presented by the italic letter.
Response 2: The gene names were made italic throughout the article.
Point 3: The formats including letter font, line thickness, and space between the lines in Tables 1 and 2 should be revised.
Response 3: A homogeneous formatting style was applied to the contents of both tables.
Point 4: Full description is required for the abbreviated terminologies that appear first time: eg., SBP (line 66), SNP (line 122), and NIH (line 270).
Response 4: All abbreviations have been spelled out in full the first time they were used in the manuscript
Point 5: In the last part of the “Discussion” section, the summarized conclusions including the significance and key points of this review are needed.
Response 5: A final paragraph was added summarizing the scope, findings, and importance of our systematic review.